# Beyond the Five Freedoms: Animal Welfare at Modern Zoological Facilities

**DOI:** 10.3390/ani13111818

**Published:** 2023-05-31

**Authors:** Lance J. Miller, Sathya K. Chinnadurai

**Affiliations:** Chicago Zoological Society—Brookfield Zoo, Brookfield, IL 60513, USA; sathya.chinnadurai@czs.org

**Keywords:** animal welfare program, program framework, staff training, animal welfare committee, animal welfare monitoring

## Abstract

**Simple Summary:**

Modern zoological facilities have an ethical responsibility to focus on animal welfare. This includes a commitment to the continuous improvement to the care of the animals as well as responding to animal welfare concerns. A modern zoological facility should have an animal welfare program that includes both proactive and reactive measures in place to focus on animal welfare. Examples include adequate staff training, an animal welfare committee, and an assessment process that monitors every individual in the collection. Improved scientific understanding, institutional standards, and public opinion have all pushed the field of zoo animal welfare forward beyond simply maintaining the Five Freedoms and instead focused on opportunities for animals to thrive in the managed environment.

**Abstract:**

The current manuscript highlights the aspects of an animal welfare program for a modern zoological facility. The program should be proactive to identify areas for continuous improvement as well as reactive to address any identified animal welfare concerns. The program should go beyond the five freedoms and utilize one of the more modern frameworks as a foundation for the program. The program should have an animal welfare committee where staff can submit animal welfare concerns without fear of retaliation. Ongoing monitoring of all individual animals should utilize both positive and negative indicators of welfare. Staff should be trained on the most current science and be able to understand key concepts about animal welfare. Facilities should also utilize new scientific findings to continuously improve animal care practices. Modern zoological institutions, including both zoos and aquariums, have an ethical responsibility to provide high levels of animal welfare for the animals under their professional care. Simply meeting minimum standards developed decades ago is not adequate, as animals should have the opportunity to thrive.

## 1. Introduction

Historically, the field of animal welfare focused on meeting minimum standards to prevent animals from suffering [1]. The well-recognized five freedoms are a good example and were considered the gold standard for years [2,3,4]. The freedoms were (1) freedom from hunger and thirst; (2) freedom from discomfort; (3) freedom from pain, injury, and disease; (4) freedom to express natural behavior; and (5) freedom from fear and distress. However, just meeting minimum standards does not allow for a facility to focus on continuous improvement in animal welfare. Modern zoological institutions, including both zoos and aquariums, have an ethical responsibility to provide high levels of animal welfare for the animals under their professional care. Simply meeting minimum standards developed decades ago is not adequate, and this has become apparent with increased standards by trade associations. Public interest has led to increased scrutiny of animal care in zoological institutions and resulted in improved documentation of welfare assessments and intervention [5,6].

While many modern accredited zoos and aquariums have been focusing on animal welfare for quite some time [7], in recent years additional accreditation standards focusing on animal welfare have been developed to ensure consistency amongst accredited institutions. The World Association of Zoos and Aquariums (WAZA) has created a 2023 Animal Welfare Goal. The Goal states “WAZA National and Regional Associations must have an animal welfare evaluation process in place and such a process must include specific elements approved by WAZA” and all institutional members must be compliant with the process [8]. Created in 2017, the Association of Zoos and Aquariums (AZA) has an accreditation standard that states “The institution must have a process for assessing animal welfare and wellness.” The process must be both proactive and reactive, transparent, and include staff or consultants with knowledge of animal welfare [9]. The process must also include the evaluation of significant life events (e.g., construction, transport, etc.). While the impact of accreditation standards has yet to be assessed across zoological facilities, there are examples in the literature of changes that have been made based on monitoring to improve the welfare of animals [10,11,12].

## 2. Animal Welfare Monitoring Program

A modern zoo or aquarium animal welfare monitoring program should be developed to focus on the continuous improvement of the welfare of the animals under their professional care. The program should include a holistic science-based approach with the goal of assessing and optimizing welfare from an animal-centric perspective [10,13]. Additionally, the program should evolve as additional scientific findings provide more information about the welfare of animals. Ideally, a modern zoological facility should have staff routinely monitoring the welfare of their animals utilizing both positive (e.g., behavioral diversity, play, exploration, etc.) and negative (e.g., stereotypic behavior, unnatural levels of aggression, etc.) indicators of animal welfare [14,15]. Absence of negative indicators of animal welfare does not demonstrate that an individual animal is thriving [14]. The welfare monitoring programs can utilize either standardized, e.g., [14], or species-specific, e.g., [16], indicators of welfare. The frequency of data collection should be based on the individual measure so that any changes detected are meaningful. For example, collecting daily weights on a bottlenose dolphin may not be practical or useful, but monitoring daily food consumption could give meaningful information [17]. Indicators should also be selected and defined so that inter- and intra-observer reliability can be maintained across all staff within a work group and/or department [18,19,20]. Reliability should be evaluated periodically to help ensure that data are meaningful. This is to ensure that despite multiple staff working the same animal, the indicators and their values are consistently identified and recorded. There is added benefit to involving staff with different roles in the assessment process. The primary animal care staff are well-equipped to detect subtle change in an animal by seeing and assessing it daily [21,22], but a manager, veterinarian, behavioral scientist, or nutritionist may bring the benefit of assessing many animals across the institution. These individuals also might not be as “close” to the animals and better able to detect changes. Comprehensive welfare assessment requires a thorough understanding of normal behavior, health, nutrition, and natural history of the species being observed.

It would be considered best practice for indicators to be closely monitored and any significant changes should be discussed during team meetings [20]. Data can also be examined on an annual basis, preferably by trained scientists if available, to look for any trends that may improve management moving forward. Management changes can be made if indicators would suggest there is a need or to focus on continuous improvement [10,13]. Management changes could include but are not limited to environmental enrichment, animal training, habitat or environmental modification, and nutrition. For all management changes, quantitative data collection should be conducted to determine the impact of the changes on animal welfare [10,13]. Objective data can include but are not limited to conducting behavioral observations, hormone analysis, or medical diagnostics.

In addition to the routine monitoring of individual animals, collection-level assessments can include a survey distributed to all animal care staff with a series of questions regarding the welfare of the animals under their professional care. Sample questions based on both the Opportunities to Thrive [14,23] and Five Domains [24,25,26] animal welfare frameworks are in Table 1.

Any concerns submitted should be evaluated to determine if there is truly an animal welfare concern. Animals that do not compare favorably to wild counterparts or professionally managed populations based on these questions may not indicate a welfare concern. For example, animals may not reproduce at similar rates due to breeding recommendations for population management purposes. Additionally, differences may exist due to other management factors (e.g., social housing, euthanasia, etc.). The main reason for comparing to both wild counterparts and professionally managed populations is that data may not be available on wild animals for all species. However, going through these questions will likely pick up on any welfare-related concerns. It is important to note the difference between animal welfare concerns and animal management concerns. Table 2 displays examples of both animal welfare and animal management concerns. While animal management concerns should also be addressed, it is important to distinguish between the two types of concerns to help prioritize efforts.

Finally, while having a robust welfare monitoring program is critical, there are certain types of events that should require an increased amount of monitoring. Termed “significant life events” [9], anytime events occur that could have a significant impact on the welfare of an animal, additional monitoring should be required. A list of example significant life events is presented in Table 3. The type and duration of increased monitoring should be determined on a case-by-case basis to focus on the welfare of the animals.

## 3. Animal Welfare Committee

A foundation to the welfare program should be an animal welfare committee with staff knowledgeable about the science of animal welfare. The number of positions and composition should be based on the needs of the facility. Some typical members might include leadership from animal care, veterinary services, and research. The chair of the animal welfare committee should likely be the head of one of those three departments. Depending on the size of the organization and the types of positions employed, staff from areas such as behavioral husbandry and nutrition are also extremely beneficial. Including individuals that are not directly involved with the animals (e.g., public relations, education, facilities and maintenance) can also provide a unique perspective. Additionally, zoos and aquariums should consider the possible benefit of adding an external non-staff member, such as someone from a local university, to help maintain objectivity.

The function or goal of the Animal Welfare Committee should be to serve as a resource for the organization. Table 4 provides a summary of some of the possible tasks and activities that can be performed by an Animal Welfare Committee at a zoo or aquarium.

## 4. Animal Welfare Concerns Process

The modern zoo or aquarium should have an animal welfare concerns process for staff to submit concerns without fear of retaliation. While ideally staff should feel comfortable submitting a concern including their name, the process should allow for anonymous concerns to be submitted prioritizing the welfare of the animals. Some potential ways to decrease fear of retaliation may include having a written policy on submitting a welfare concern, training staff on the welfare concern policy, have a written policy for investigating retaliation claims, and maintaining transparency with staff. However, when the submitted concern includes their name, this provides a great opportunity for the committee to ask questions if anything about the concern is unclear. While the animal welfare concern form can be designed to meet the needs of the facility, general information should include the option to include their name, date, species or individual of concern, evidence of the welfare concern, and any potential solutions.

The first step for the committee when receiving a concern should be to investigate in order to determine if an actual animal welfare concern exists. This can be done through a variety of methods, including but not limited to staff interviews, behavioral observations, veterinary exams, or biological sampling to examine physiology. If it is determined that no animal welfare concern exists, the individual who submitted the concern should be notified of the findings. If the concern was submitted anonymously, it would be considered best practice to inform the entire animal care staff that works directly with the species or individual animal that was identified as having a potential concern.

When an animal welfare concern has been identified, there may or may not be an apparent cause. If the cause can easily be identified (e.g., wounds from aggressive interactions due to one feeding location), then the challenge should be addressed immediately. However, if the cause of the welfare concern is not apparent, brainstorming sessions may be necessary to try and determine potential solutions. Individuals involved in the session can include staff that work directly with the species or individual animal with the concern, veterinary services, behavioral husbandry, nutrition, and research. Individuals involved in these brainstorming sessions can vary depending on the institution. Table 5 includes questions to ask during the brainstorming sessions to try and identify potential solutions. When asking these questions, always consider the natural and individual history of the species or individual. Depending on the concern, all questions may not be applicable, but sometimes thinking outside the box during the brainstorming sessions can be constructive. Once some potential solutions have been identified, research projects can be designed to examine how management changes impact animal welfare for the species or individual of concern [10,13].

## 5. Staff Training

Another critical component to a welfare monitoring program is having staff knowledgeable about animal welfare. Initial training of all animal care staff should include indicators of welfare utilized for each species, a general overview of the difference between animal welfare and animal rights, as well as the difference between animal welfare concerns and animal management concerns. It would also be recommended to have routine follow-up with staff to make sure the indicators of animal welfare are being utilized correctly along with any updated materials based on new science. When successful, all staff should be able to identify the indicators being used for their particular species and be able to bring up data from a species to demonstrate their knowledge and understanding of the program and the status of the animals under their care. As animal welfare is a science, staff should also routinely be retrained as our knowledge continues to grow with new research findings.

## 6. Animal Welfare Management

In addition to a holistic welfare monitoring program, animal welfare management should be based on current science. An animal welfare management program should draw on all areas, including but not limited to animal care and husbandry, behavioral husbandry, veterinary services, nutrition, and research. The welfare management program should be adaptive, turning recent scientific findings into practice.

## 7. Conclusions

Advancement in the science and practice of wild animal welfare have led to improved standards and documentation of animal welfare in modern zoos and aquariums. Improved scientific understanding, institutional standards, and public opinion have all pushed the field of zoo animal welfare forward beyond simply maintaining the Five Freedoms to instead focus on opportunities for animals to thrive in the managed environment.

## Figures and Tables

**Table 1 animals-13-01818-t001:** Sample questions to proactively survey staff about potential animal welfare concerns.

Animal Welfare Assessment Questions
Do the animals under your professional care reproduce at rates equal or greater than their wild counterparts or professionally managed populations?
2. Do the animals under your professional care have mortality rates higher than their wild counterparts or professionally managed populations?
3. Do the animals under your professional care experience chronic disease more than their wild counterparts or professionally managed populations?
4. Do the animals under your professional care exhibit stereotypic or abnormal behavior?
5. Do the animals under your professional care exhibit behavioral diversity at levels equal or greater than their wild counterparts or professionally managed populations?
6. Do the animals under your professional care explore and use their entire environment?
7. Do the animals under your professional care display social behaviors at a level similar to their wild counterparts?

**Table 2 animals-13-01818-t002:** Examples of animal welfare and animal management concerns.

Sample Animal Welfare Concerns	Sample Animal Management Concerns
The stage for the nighttime concert was placed next to the dolphin habitat. The day following the concert, the animals would not eat their entire diet or participate in training sessions.	In the bottlenose dolphin habitat, there is a sharp edge on one of the habitat walls. The edge needs to be fixed before one of the animals injures themselves.
2. The new species of macaw at the zoo engages in feather plucking approximately 12% of the time.	2. We do not have any information on appropriate diets, social groupings, or species-appropriate behavior for the new species of macaw we are bringing in next month.
3. Swimming in tight circles in the small habitat has led to spinal deformities in our sharks.	3. We do not have an extra habitat for our sharks to perform maintenance on the current exhibit.
4. The infant mortality rate for our flamingos is around 15%, which is much higher than other zoological facilities and the wild.	4. Our flamingo habitat is a dump-and-fill system instead of a filtration system. The system should be installed to continuously keep the water clean.

**Table 3 animals-13-01818-t003:** Potential significant life events that call for increased welfare monitoring.

Example Significant Life Events
***Animal Introductions***—Changes in social groupings that are not routine or the addition of any new individual animals.
***Animal Shipment***—An animal is scheduled to be shipped to a zoo or aquarium.
***Births/Deaths***—Changes in group composition due to the birth or death of an individual.
***Construction***—Any construction that will cause increased noise levels above 110 dB 30 feet from the source of the construction or vibrations for animals closest to the site. Please note that dB level and frequency may vary depending on species.
***Events (i.e., Concerts, Weddings, Festivals, etc.)***—Any new events that are significantly different than previous events held at the facility for the animals closet to the location of the event.
***Habitat Change***—Anytime an animal moves from one habitat into another.
***Hospitalization***—Anytime an animal is hospitalized due to injury or illness.
***Medical Procedures***—Anytime an animal undergoes a medical procedure for injury or illness.
***Quarantine***—Anytime an animal is in quarantine following shipment.
***Quarantine Release***—Anytime an animal moves from quarantine into the collection.
***Significant Diet Change***—Any significant change in the quantity or type of diet items.
***Staff Changes***—Any staff changes where species are known to have strong human–animal relationships.

**Table 4 animals-13-01818-t004:** Summary table of potential tasks/activities for the animal welfare committee.

Task/Activity	Description
Animal Welfare Concerns Process	Staff at an organization should be aware of the process in place for them to submit an animal welfare concern without fear of retaliation. This process can be led by an Animal Welfare Committee.
Staff Training	All staff should be trained and well-versed in the science of animal welfare and how to report any animal welfare concerns.
Development of Animal Welfare Assessment Process	The Animal Welfare Committee can work with each of the teams to assist in developing a robust animal welfare assessment.
Animal Welfare Assessment Data Review	The Animal Welfare Committee or a subcommittee can review assessment data to look for patterns or trends.

**Table 5 animals-13-01818-t005:** Sample list of questions to ask when brainstorming potential solutions to an animal welfare concern.

Sample Brainstorming Questions
Are there any behavioral needs that are currently not being met?
2. Are there any potential nutritional deficiencies or concerns related to diet?
3. Are there any environmental conditions (e.g., humidity, temperature, noise, etc.) that could be causing the concern?
4. Are any aspects of the habitat (e.g., concrete, feeding locations, etc.) that could be causing the concern?
5. Is group composition appropriate for the species?
6. Are there any medical conditions that could be causing the concern?
7. What opportunities for choice and control are provided?
8. What opportunities to self-maintain are provided?
9. What can the species or individual see outside of their habitat?
10. Are there any potential visitor effects that may be causing the concern?
11. If the habitat has multiple species, could that be causing the concern?
12. Have any significant life events occurred recently?

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
