# Peer review of "Beyond the Five Freedoms: Animal Welfare at Modern Zoological Facilities"

_animals, 2023, doi:10.3390/ani13111818_

Round 1
Reviewer 1 Report
Informative and clear commentary on an important topic. Authors give great advice on how zoos may achieve increased animal welfare standards. The piece reads well, I just have a few small suggestions/tweaks:
Line 67-68: Behavioural diversity is debated as an effective method to evaluate positive behaviours, add (an) additional example(s) of positive behavioural indicators.
Line 72: Confusing, change to: 'so that any changes detected are meaningful'
Lines 81-82. Manager, veterinarian or nutritionist may also notice changes over a longer time period as they are not as 'close' to the animal.
Table 1: I do not agree with comparatives to the wild here. Captivity provides many different pressures that are not comparable. For example:
- Reproduction, animals might not be part of a breeding program or given recommendations to breed;
- Mortality rates - animals might be euthanised;
- Behavioural diversity - captive animals will not express the same levels of behavioural diversity as wild animals due to the different needs and provisions. Additionally, is there always wild data available to compare for this?
-Social behaviours - captive animals may not be housed in similar social groupings due to space/captive pressures. Additionally, some species are not housed as per their wild counterparts (tigers, orangutan) and it is suggested that housing them in groups rather than alone is better for welfare but this will change their social behaviour compared to the wild.
Table 2: This is great and I am sure many will find this really useful.
Lines 188-126: I feel that at least one member of the AWC should be external to the zoo. This will help maintain an objective, non-bias focus. Examples could include suitably qualified/trained University staff/researchers or staff from animal welfare charities.
Table 4: Staff training - is this ALL staff - including retail/food relevant staff or just animal-linked staff? Important to make this clear.
General: Tables align left rather than centred for ease of reading (depending on Animals formatting).
Author Response
File attached

Reviewer 2 Report
This commentary is a really nice and concise summary for future of animal welfare monitoring in the zoo industry. It follows the standards set by WAZA as part of the 2023 Welfare Goals and is written in an accessible way that can be appreciated by managers, scientists and keepers.
I have 3 very minor comments for the authors which I've listed below but I'm very happy to accept this paper for publication as it will provide a much needed validation to the upcoming changes in animal welfare assessment!
Line 81 - you could include behaviour scientist in the list alongside nutritionist etc. Behaviourists are mentioned in subsequent lists in the document so I believe this would fit well.
Table 1, Q1 - could there be a note related to this as breeding success may not represent a free mating system and maybe dictated by the studbook holder? I appreciate that excessive breeding could result in a welfare issue but perhaps this is a nice example as you go on to describe the difference between welfare and management issues.
Table 3 Construction - you could include vibration as well as dB levels here as we know animals can be very sensitive to this.
Author Response
File attached

Reviewer 3 Report
The current manuscript provides recommendations for the inclusion of an animal welfare program for a modern zoological facility. It is well written and provides great content about how to be proactive and reactive to the needs of animals in collections, but also presents the need for continual improvements to be made to ensure that animal welfare is monitored and maintained. I argue if an Animal Welfare Committee would be a safe place for zoo staff to go to share their concerns about animal welfare, as I know the authors will appreciate that whistleblowing is a contentious issue in zoos. Here are my more specific comments:
Line 41: Can you please briefly expand on the ‘standards’ suggested here.
Line 61: Do you mean ‘monitoring’ rather than ‘mentoring’?
Line 69-70: Commendation to the authors on this very important statement!
Line 75: Please add a hyphen to the ‘inter’.
Lines 75-78: How can you ensure that multiple members of staff working with the same animal, or groups of animals, are consistently recording data in the same way? It might be useful to see another recommendation for this here, such as standardisation, across experience and less experienced staff.
Line 85-93: What would you do with this data after it have been collected? There is only mention of ‘collecting’ the data, but what actually needs to be done with the data? Would it be analysed against previous data collected for example? Would you need to ensure you have a scientist working for, or with, the zoo?
Lines 99-100: Table 1 - I would like to see a discussion around the questions in this table. It is great to see this included but there could be greater discussion around whether the rates of reproduction, equal to wild counterparts, are actually a measure of welfare; good or bad. Please consider adding in more detail/discussion around the evolution of these questions.
Line 116: Table 3 - I feel there should a be a note added to the section about ‘Construction’ noise as the dB levels will vary between species. Again, a note for the ‘Events’ section, as the event may have happened previously, but timescale between these same events may affect the expressed behaviour/potential habituation to the disruption the event may have caused.
Lines 118-126: This is an easy-to-achieve recommendation and it is sad to see that so few organisations have this currently in place. I hope this recommendation is taken on by the majority of collections.
Line 130: Table 4 – I think there should be a note here about fear of retaliation here as whistleblowing in the zoo world often leads to job losses. How would you mitigate this? How could you encourage an open and environment for sharing welfare concerns?
Line 152: remove comma after the e.g.
Lines 176-179: I would possibly also add in here about having a regular cadence of ensuring this training is up to date. With new science coming out all the time, it would be useful to ensure that there is someone on staff who is able to ensure the latest science/welfare information, per species, is shared with the relevant staff. This comment also applies to section 6.
Overall, this is a great addition to the literature around zoo and aquarium animal welfare. It is well-written and could be shared with relevant zoo personnel in its current form. I would encourage the authors to find a way to get the recommendations from this paper into the hands of those working in zoos and aquariums, to increase the uptake of these realistic and achievable recommendations.
Author Response
File attached
